# WHAT IS MISSING IN IRM TRAINING AND EVALUATION? CHALLENGES AND SOLUTIONS

**Yihua Zhang**[1], **Pranay Sharma**[2], **Parikshit Ram**[3], **Mingyi Hong**[4], **Kush Varshney**[3], **Sijia Liu**[1,3]
[1]Michigan State University, [2]Carnegie Mellon University, [3]IBM Research, [4]University of Minnesota

## ABSTRACT

Invariant risk minimization (IRM) has received increasing attention as a way to acquire environment-agnostic data representations and predictions, and as a principled solution for preventing spurious correlations from being learned and for improving models' out-of-distribution generalization. Yet, recent works have found that the optimality of the originally-proposed IRM optimization (IRMv1) may be compromised in practice or could be impossible to achieve in some scenarios. Therefore, a series of advanced IRM algorithms have been developed that show practical improvement over IRMv1. In this work, we revisit these recent IRM advancements, and identify and resolve three practical limitations in IRM training and evaluation. First, we find that the effect of batch size during training has been chronically overlooked in previous studies, leaving room for further improvement. We propose small-batch training and highlight the improvements over a set of large-batch optimization techniques. Second, we find that improper selection of evaluation environments could give a false sense of invariance for IRM. To alleviate this effect, we leverage diversified test-time environments to precisely characterize the invariance of IRM when applied in practice. Third, we revisit Ahuja et al. (2020)'s proposal to convert IRM into an ensemble game and identify a limitation when a single invariant predictor is desired instead of an ensemble of individual predictors. We propose a new IRM variant to address this limitation based on a novel viewpoint of ensemble IRM games as consensus-constrained bi-level optimization. Lastly, we conduct extensive experiments (covering 7 existing IRM variants and 7 datasets) to justify the practical significance of revisiting IRM training and evaluation in a principled manner.

## 1 INTRODUCTION

Deep neural networks (DNNs) have enjoyed unprecedented success in many real-world applications (He et al., 2016; Krizhevsky et al., 2017; Simonyan & Zisserman, 2014; Sun et al., 2014). However, experimental evidence (Beery et al., 2018; De Haan et al., 2019; DeGrave et al., 2021; Geirhos et al., 2020; Zhang et al., 2022b) suggests that DNNs trained with empirical risk minimization (**ERM**), the most commonly used training method, are prone to reproducing spurious correlations in the training data (Beery et al., 2018; Sagawa et al., 2020). This phenomenon causes performance degradation when facing distributional shifts at test time (Gulrajani & Lopez-Paz, 2020; Koh et al., 2021; Wang et al., 2022; Zhou et al., 2022a). In response, the problem of *invariant prediction* arises to enforce the model trainer to learn stable and causal features (Beery et al., 2018; Sagawa et al., 2020).

In pursuit of out-of-distribution generalization, a new model training paradigm, termed invariant risk minimization (**IRM**) (Arjovsky et al., 2019), has received increasing attention to overcome the shortcomings of ERM against distribution shifts. In contrast to ERM, IRM aims to learn a universal representation extractor, which can elicit an *invariant* predictor across multiple training environments. However, different from ERM, the learning objective of IRM is highly non-trivial to optimize in practice. Specifically, IRM requires solving a challenging bi-level optimization (**BLO**) problem with a hierarchical learning structure: invariant representation learning at the *upper-level* and invariant predictive modeling at the *lower-level*. Various techniques have been developed to solve IRM effectively, such as (Ahuja et al., 2020; Lin et al., 2022; Rame et al., 2022; Zhou et al., 2022b) to name a few. Despite the proliferation of IRM advancements, several issues in the theory and practice have also appeared. For example, recent works (Rosenfeld et al., 2020; Kamath et al.,

2021) revealed the theoretical failure of IRM in some cases. In particular, there exist scenarios where the optimal invariant predictor is impossible to achieve, and the IRM performance may fall behind even that of ERM. Practical studies also demonstrate that the performance of IRM rely on multiple factors, *e.g.*, model size (Lin et al., 2022; Zhou et al., 2022b), environment difficulty (Dranker et al., 2021; Krueger et al., 2021), and dataset type (Gulrajani & Lopez-Paz, 2020).

Therefore, key challenges remain in deploying IRM to real-world applications. In this work, we revisit recent IRM advancements and uncover and tackle several pitfalls in IRM training and evaluation, which have so far gone overlooked. We first identify the large-batch training issue in existing IRM algorithms, which prevents escape from bad local optima during IRM training. Next, we show that evaluation of IRM performance with a *single* test-time environment could lead to an inaccurate assessment of prediction invariance, even if this test environment differs significantly from training environments. Based on the above findings, we further develop a novel IRM variant, termed **BLOC**-IRM, by interpreting and advancing the IRM-GAME method (Ahuja et al., 2020) through the lens of **BLO** with **C**onsensus prediction. Below, we list our **contributions** (❶-❹).

❶ We demonstrate that the prevalent use of large-batch training leaves significant room for performance improvement in IRM, something chronically overlooked in the previous IRM studies with benchmark datasets COLORED-MNIST and COLORED-FMNIST. By reviewing and comparing with 7 state-of-the-art (SOTA) IRM variants (Table 1), we show that simply using *small-batch* training improves generalization over a series of more involved large-batch optimization enhancements.

❷ We also show that an inappropriate evaluation metric could give a false sense of invariance to IRM. Thus, we propose an extended evaluation scheme that quantifies both precision and 'invariance' across diverse testing environments.

❸ Further, we revisit and advance the IRM-GAME approach (Ahuja et al., 2020) through the lens of consensus-constrained BLO. We remove the need for an ensemble (one per training environment) of predictors in IRM-GAME by proposing **BLOC-IRM** (**BLO** with **C**onsensus **IRM**), which produces a single invariant predictor.

❹ Lastly, we conduct extensive experiments (on 7 datasets, using diverse model architectures and training environments) to justify the practical significance of our findings and methods. Notably, we conduct experiments on the CELEBA dataset as a new IRM benchmark with realistic spurious correlations. We show that BLOC-IRM outperforms all baselines in nearly all settings.

## 1.1 RELATED WORK

**IRM methods.** Inspired by the invariance principle (Peters et al., 2016), Arjovsky et al. (2019) define IRM as a BLO problem, and develop a relaxed single-level formulation, termed IRMV1, for ease of training. Recently, there has been considerable work to advance IRM techniques. Examples of IRM variants include penalization on the variance of risks or loss gradients across training environments (Chang et al., 2020; Krueger et al., 2021; Rame et al., 2022; Xie et al., 2020; Xu & Jaakkola, 2021; Xu et al., 2022), domain regret minimization (Jin et al., 2020), robust optimization over multiple domains (Xu & Jaakkola, 2021), sparsity-promoting invariant learning (Zhou et al., 2022b), Bayesian inference-baked IRM (Lin et al., 2022), and ensemble game over the environment-specific predictors (Ahuja et al., 2020). We refer readers to Section 2 and Table 1 for more details on the IRM methods that we will focus on in this work.

Despite the potential and popularity of IRM, some works have also shown the theoretical and practical limitations of current IRM algorithms. Specifically, Chen et al. (2022); Kamath et al. (2021) show that invariance learning via IRM could fail and be worse than ERM in some two-bit environment setups on COLORED-MNIST, a synthetic benchmark dataset often used in IRM works. The existence of failure cases of IRM is also theoretically shown by Rosenfeld et al. (2020) for both linear and non-linear models. Although subsequent IRM algorithms take these failure cases into account, there still exist huge gaps between theoretically desired IRM and its practical variants. For example, Lin et al. (2021; 2022); Zhou et al. (2022b) found many IRM variants incapable of maintaining graceful generalization on large and deep models. Moreover, Ahuja et al. (2021); Dranker et al. (2021) demonstrated that the performance of IRM algorithms could depend on practical details, *e.g.*, dataset size, sample efficiency, and environmental bias strength. The above IRM limitations in-

spire our work to study when and how we can turn the IRM advancements into effective solutions, to gain high-accuracy and stable invariant predictions in practical scenarios.

**Domain generalization.** IRM is also closely related to domain generalization (Carlucci et al., 2019; Gulrajani & Lopez-Paz, 2020; Koh et al., 2021; Li et al., 2019; Nam et al., 2021; Wang et al., 2022; Zhou et al., 2022a). Compared to IRM, domain generalization includes a wider range of approaches to improve prediction accuracy against distributional shifts (Beery et al., 2018; Jean et al., 2016; Koh et al., 2021). For example, an important line of research is to improve representation learning by encouraging cross-domain feature resemblance (Long et al., 2015; Tzeng et al., 2014). The studies on domain generalization have also been conducted across different learning paradigms, *e.g.*, adversarial learning (Ganin et al., 2016), self-supervised learning (Carlucci et al., 2019), and meta-learning (Balaji et al., 2018; Dou et al., 2019).

## 2 PRELIMINARIES AND SETUP

In this section, we introduce the basics of IRM and provide an overview of our IRM case study.

**IRM formulation.** In the original IRM framework Arjovsky et al. (2019), consider a supervised learning paradigm, with datasets $\{\mathcal{D}^{(e)}\}_{e \in \mathcal{E}_{\mathrm{tr}}}$ collected from $N$ training environments $\mathcal{E}_{\mathrm{tr}} = \{1, 2, \ldots, N\}$. The training samples in $\mathcal{D}^{(e)}$ (corresponding to the environment $e$) are of the form $(\mathbf{x}, y) \in \mathcal{X} \times \mathcal{Y}$, where $\mathcal{X}$ and $\mathcal{Y}$ are, respectively, the raw feature space and the label space. IRM aims to find an *environment-agnostic data representation $\phi_{\boldsymbol{\theta}} : \mathcal{X} \to \mathcal{Z}$*, which elicits an *invariant prediction $f_{\mathbf{w}} : \mathcal{Z} \to \mathcal{Y}$* that is simultaneously optimal for all environments. Here $\boldsymbol{\theta}$ and $\mathbf{w}$ denote model parameters to be learned, and $\mathcal{Z}$ denotes the representation space. Thus, IRM yields an *invariant predictor $f_{\mathbf{w}} \circ \phi_{\boldsymbol{\theta}} : \mathcal{X} \to \mathcal{Y}$* that can generalize to unseen test-time environments $\{\mathcal{D}^{(e)}\}_{e \notin \mathcal{E}_{\mathrm{tr}}}$. Here $\circ$ denotes function composition, *i.e.*, $f_{\mathbf{w}} \circ \phi_{\boldsymbol{\theta}}(\cdot) = f_{\mathbf{w}}(\phi_{\boldsymbol{\theta}}(\cdot))$. We will use $\mathbf{w} \circ \boldsymbol{\theta}$ as a shorthand for $f_{\mathbf{w}} \circ \phi_{\boldsymbol{\theta}}$. IRM constitutes the following BLO problem:

$$\underset{\boldsymbol{\theta}}{\text{minimize}} \quad \sum_{e \in \mathcal{E}_{\mathrm{tr}}} \ell^{(e)}(\mathbf{w}^*(\boldsymbol{\theta}) \circ \boldsymbol{\theta}); \quad \text{subject to} \quad \mathbf{w}^*(\boldsymbol{\theta}) \in \arg\min_{\bar{\mathbf{w}}} \ell^{(e)}(\bar{\mathbf{w}} \circ \boldsymbol{\theta}), \ \forall e \in \mathcal{E}_{\mathrm{tr}}, \quad \text{(IRM)}$$

where $\ell^{(e)}(\mathbf{w} \circ \boldsymbol{\theta})$ is the per-environment training loss of the predictor $\mathbf{w} \circ \boldsymbol{\theta}$ under $\mathcal{D}^{(e)}$. Clearly, IRM involves two optimization levels that are coupled through the lower-level solution $\mathbf{w}^*(\boldsymbol{\theta})$. Achieving the desired invariant prediction requires the solution sets of the individual lower-level problems $\{\arg\min_{\bar{\mathbf{w}}} \ell^{(e)}(\bar{\mathbf{w}} \circ \boldsymbol{\theta}), e \in \mathcal{E}_{tr}\}$ to be non-singleton. However, BLO problems with non-singleton lower-level solution sets are significantly more challenging (Liu et al., 2021). To circumvent this difficulty, Arjovsky et al. (2019) relax (IRM) into a single-level optimization problem (*a.k.a.*, IRMv1):

$$\underset{\boldsymbol{\theta}}{\text{minimize}} \quad \sum_{e \in \mathcal{E}_{\mathrm{tr}}} [\ell^{(e)}(\boldsymbol{\theta}) + \gamma \|\nabla_{w|w=1.0} \ell^{(e)}(w \circ \boldsymbol{\theta})\|_2^2], \quad \text{(IRMv1)}$$

where $\gamma > 0$ is a regularization parameter and $\nabla_{w|w=1.0} \ell^{(e)}$ denotes the gradient of $\ell^{(e)}$ with respect to $w$, computed at $w = 1.0$. Compared with IRM, IRMv1 is restricted to *linear* invariant predictors, and penalizes the deviation of individual environment losses from stationarity to approach the lower-level optimality in (IRM). IRMv1 uses the fact that a scalar predictor ($w = 1.0$) is equivalent to a linear predictor. Despite the practical simplicity of (IRMv1), it may fail to achieve the desired invariance (Chen et al., 2022; Kamath et al., 2021).

**Case study of IRM methods.** As illustrated above, the objective of IRM is difficult to optimize, while IRMv1 only provides a sub-optimal solution. Subsequent advances have attempted to reduce this gap. In this work, we focus on 7 popular IRM variants and evaluate their invariant prediction performance over 7 datasets. **Table 1** and **Table 2** respectively summarize the IRM methods and the datasets considered in this work. We survey the most representative and effective IRM variants in the literature, which will also serve as our baselines in performance comparison.

Following Table 1, we first introduce the **IRMV0** variant, a generalization of IRMv1, by relaxing its assumption of linearity of the predictor $\mathbf{w}$, yielding

$$\underset{\mathbf{w}, \boldsymbol{\theta}}{\text{minimize}} \quad \sum_{e \in \mathcal{E}_{\mathrm{tr}}} [\ell^{(e)}(\mathbf{w} \circ \boldsymbol{\theta}) + \gamma \|\nabla_{\mathbf{w}} \ell^{(e)}(\mathbf{w} \circ \boldsymbol{\theta})\|_2^2]. \quad \text{(IRMv0)}$$

Next, we consider the risk extrapolation method **REx** (Krueger et al., 2021), an important baseline based on distributionally robust optimization for group shifts (Sagawa et al., 2019). Furthermore,

Table 1: Summary of the 7 existing IRM variants considered in this work, and the proposed BLOC-IRM method (see Section 5). We also list the 7 benchmark datasets used to evaluate IRM performance, namely, COLORED-MNIST (CoM), COLORED-FMNIST (CoF), CIFAR-MNIST (CiM), COLORED-OBJECT (CoO), CELEBA (CA), PACS (P) and VLCS (A). The symbols '✓' signifies the dataset used in the specific reference.

| IRM Method | Venue | Datasets | | | | | | | Reference |
|---|---|---|---|---|---|---|---|---|---|
| | | CoM | CoF | CiM | CoO | CA | P | V | |
| IRMV1 | arXiv | ✓ | | | | | | | (Arjovsky et al., 2019) |
| IRMV0 | N/A | | | | | | | | This Work |
| IRM-GAME | ICML | ✓ | ✓ | | | | | | (Ahuja et al., 2020) |
| REX | ICML | ✓ | ✓ | | | | ✓ | ✓ | (Krueger et al., 2021) |
| BIRM | CVPR | ✓ | | ✓ | ✓ | | | | (Lin et al., 2022) |
| SPARSEIRM | ICML | ✓ | | ✓ | ✓ | | | | (Zhou et al., 2022b) |
| FISHR | ICML | ✓ | | | | | ✓ | ✓ | (Rame et al., 2022) |
| Ours | N/A | ✓ | ✓ | ✓ | ✓ | ✓ | ✓ | ✓ | This Work |

Table 2: Dataset setups. 'Invariant' and 'Spurious' represent the core and spurious features. 'Env1' and 'Env2' are environments with different spurious correlations.

| Dataset | Invariant | Spurious | Env 1 | Env 2 |
|---|---|---|---|---|
| CoM | Digit | Color | | |
| CoF | Object | Color | | |
| CiM | CIFAR | MNIST | | |
| CoO | Object | Color | | |
| CA | Smiling | Hair Color | | |
| P | Object | Texture | | |
| V | Object | Environment | | |

inspired by the empirical findings that the performance of IRM could be sensitive to model size (Choe et al., 2020; Gulrajani & Lopez-Paz, 2020), we choose the SOTA methods Bayesian IRM (**BIRM**) (Lin et al., 2022) and sparse IRM (**SPARSEIRM**) (Zhou et al., 2022b), both of which show improved performance with large models. Also, we consider the SOTA method **FISHR** (Rame et al., 2022), which modifies IRM to penalize the domain-level gradient variance in single-level risk minimization. **FISHR** provably matches both domain-level risks and Hessians. Lastly, we include **IRM-GAME** (Ahuja et al., 2020) as a special variant of IRM. Different from the other methods which seek an invariant predictor, IRM-GAME endows each environment with a predictor, and leverages this *ensemble* of predictors to achieve invariant representation learning. This is in contrast to other existing works which seek an invariant predictor. Yet, we show in Section 5 that IRM-GAME can be interpreted through the lens of consensus-constrained BLO and generalized for invariant prediction. We also highlight that diverse dataset types are considered in this work (see Table 2) to benchmark IRM's performance. More details on dataset selections can be found in Appendix A.

## 3 LARGE-BATCH TRAINING CHALLENGE AND IMPROVEMENT

In this section, we demonstrate and resolve the large-batch training challenge in current IRM implementations (Table 1).

**Large-batch optimization causes instabilities of IRM training.** Using very large-size batches for model training can result in the model getting trapped near a bad local optima (Keskar et al., 2016). This happens as a result of the lack of stochasticity in the training process, and is known to exist even in the ERM paradigm (Goyal et al., 2017; You et al., 2017a). Yet, nearly all the existing IRM methods follow the training setup of IRMV1 (Arjovsky et al., 2019), which used the *full-batch* gradient descent (GD) method rather than the *mini-batch* stochastic gradient descent (SGD) for IRM training over COLORED-MNIST and COLORED-FMNIST. In the following, we show that large-batch training might give a false impression of the relative ranking of IRM performances.

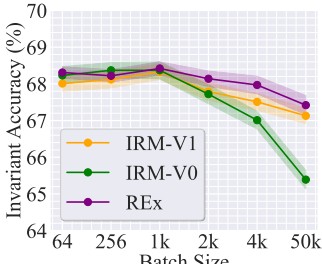

Figure 1: The performance of three IRM methods (IRMV1, IRMV0, and REX) vs. batch-size under COLORED-MNIST. The full batch-size is 50k.

We start with an exploration of the impact of batch size on the invariant prediction accuracy of existing IRM methods under COLORED-MNIST. Here the invariant prediction accuracy refers to the averaged accuracy of the invariant predictor applied to diverse test-time environments. We defer its formal description to Section 4. **Figure 1** shows the invariant prediction accuracy of three IRM methods IRMV1, IRMV0, and REX vs. the data batch size (see Figure A1 for results of other IRM variants and Figure A5 for COLORED-FMNIST). Recall that the full batch size (50k) was used in the existing IRM implementations (Arjovsky et al., 2019; Krueger et al., 2021). As we can see, in the full-batch setup, IRM methods lead to widely different invariant prediction accuracies, where REX and IRMV1 significantly outperform IRMV0. In contrast, in the small-batch case (with size

1k), the discrepancy in accuracy across methods vanishes. We see that IRMv0 can be as effective as IRMv1 and other IRM variants (such as REx) *only if* an appropriate *small* batch size is used.

Empirical evidence in Figure 1 shows that large-batch IRM training is less effective than small-batch. This is aligned with the observations in ERM (You et al., 2017b; 2018; 2019), where the lack of stochasticity makes the optimizer difficult to escape from a sharp local minimum. We also justify this issue by visualizing the loss landscapes in Figure A2. Notably, the small-batch training enables IRMv1 to converge to a local optimum with a flat loss landscape, indicating better generalization (Keskar et al., 2016).

**Small-batch training is effective versus a zoo of large-batch optimization enhancements.** To mitigate the large-batch IRM training issue, we next investigate the effectiveness of both small-batch training and a zoo of large-batch optimization enhancements. Inspired by large-batch training techniques to scale up ERM, we consider Large-batch SGD (**LSGD**) (Goyal et al., 2017) and Layer-wise Adaptive Learning Rate (**LALR**) (You et al., 2017b; 2018; 2019; Zhang et al., 2022a). Both methods aim to smoothen the optimization trajectory by improving either the learning rate scheduler or the quality of initialization. Furthermore, we adopt sharpness-aware minimization (**SAM**) (Foret et al., 2020) as another possible large-batch training solution to explicitly penalize the sharpness of the loss landscape. We integrate the above optimization techniques with IRM, leading to the variants **IRM-LSGD**, **IRM-LALR**, and **IRM-SAM**. See Appendix B.1 for more details.

In Table 3, we compare the performance of the simplest small-batch IRM training with that of those large-batch optimization technique-integrated IRM variants (*i.e.*, 'LSGD/LALR/SAM' in the Table). As we can see, the use of large-batch optimization techniques indeed improves the prediction accuracy over the original IRM implementation. We also observe that the use of SAM for IRM is consistently better than LALR and LSGD, indicating the promise of SAM to scale up IRM with a large batch size. Yet, the small-batch training protocol consistently outperforms large-batch training across all the IRM variants (see the column 'Small'). Additional experiment results in Section 6 show that small-batch IRM training is effective across datasets, and promotes the invariance achieved by all methods.

Table 3: Prediction accuracy of IRM methods on COLORED-MNIST using the original large-batch implementation ('Original'), the large-batch optimization-integrated implementations ('LSGD/LALR/SAM'), and the small-batch training recipe ('Small').

| Method | Original | LSGD | LALR | SAM | **Small** |
|---|---|---|---|---|---|
| IRMv1 | 67.13 | 67.31 | 67.44 | 67.79 | **68.33** |
| IRMv0 | 65.39 | 66.42 | 66.76 | 66.99 | **68.37** |
| IRM-GAME | 65.69 | 65.82 | 65.47 | 66.23 | **67.73** |
| REx | 67.42 | 67.53 | 67.59 | 67.82 | **68.42** |
| BIRM | 67.93 | 67.99 | 68.21 | 68.32 | **68.71** |
| SPARSEIRM | 67.72 | 67.85 | 67.99 | 68.13 | **68.81** |
| FISHR | 67.88 | 67.82 | 67.93 | 68.11 | **68.69** |
| Average | 67.02 | 67.25 | 67.34 | 67.63 | **68.44** |

## 4 MULTI-ENVIRONMENT INVARIANCE EVALUATION

In this section, we revisit the evaluation metric used in existing IRM methods, and show that expanding the diversity of test-time environments would improve the accuracy of invariance assessment.

Nearly all the existing IRM methods (including those listed in Table 1) follow the evaluation pipeline used in the vanilla IRM framework (Arjovsky et al., 2019), which assesses the performance of the learned invariant predictor on a single unseen test environment. This test-time environment is significantly different from train-time environments. For example, COLORED-MNIST (Arjovsky et al., 2019) suggests a principled way to define two-bit environments, widely-used for IRM dataset curation. Specifically, the COLORED-MNIST task is to predict the label of the handwritten digit groups (digits 0-4 for group 1 and digits 5-9 for group 2). The digit number is also spuriously correlated with the digit color (Table 2). This spurious correlation is controlled by an *environment bias parameter* $\beta$, which specifies different data environments with different levels of spurious correlation[1]. In (Arjovsky et al., 2019), $\beta = 0.1$ and $\beta = 0.2$ are used to define *two training environments*, which sample the color ID by flipping the digit group label with probability $10\%$ and $20\%$, respectively. At *test time*, the invariant accuracy is evaluated on a single, unseen environment with $\beta = 0.9$.

However, the prediction accuracy of IRM could be sensitive to the choice of test-time environment (*i.e.*, the value of $\beta$). For the default test environment $\beta = 0.9$, the predictor performance of three representative IRM methods (IRMv1, IRM-GAME, FISHR) ranked from high to low is IRM-GAME>FISHR>IRMv1. Given this apparent ranking, we explore more diverse test-time environments, generated by $\beta \in \Omega := \{0.05, 0.1, \ldots, 0.95\}$.

---

[1]In the two-bit environment, there exists another environment parameter $\alpha$ that controls the label noise level.

Although the train-time bias parameters $\{0.1, 0.2\}$ belong to $\Omega$, test data is generated afresh, different from training data. We see in Figure 2A that the superiority of IRM-GAME at $\beta = 0.9$ vanishes for smaller $\beta$. Consequently, for invariant prediction evaluated in other testing environments (e.g., $\beta < 0.4$), the performance ranking of the same methods becomes IRMV1>FISHR>IRM-GAME. This mismatch of results suggests we measure the 'invariance' of IRM methods against diverse test environments. Otherwise, evaluation with single $\beta$ could give a false sense of invariance. In Figure 2B, we present the box plots of prediction accuracies for IRM variants, over the diverse set of testing environments ($\beta \in \Omega$). Evidently, IRMV1, the oldest

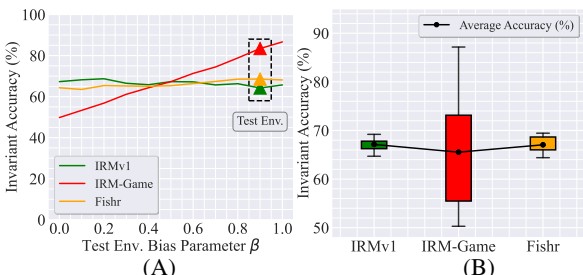

Figure 2: Performance comparison of IRM variants IRMV1, IRM-GAME, and FISHR on COLORED-MNIST. (A) Evaluation in different test-time environments (corresponding to different $\beta$). $\beta$ values used by the two training environments are $0.1, 0.2$ respectively. The conventional evaluation is done with the test environment $\beta = 0.9$ (see '▲'). (B) Box plots of prediction accuracies over diverse test environments corresponding to $\beta \in \{0.05, 0.1, \dots, 0.95\}$. IRMV1 achieves the best average accuracy (67.13%), followed by FISHR (67.05%) and IRM-GAME (65.53%).

(sub-optimal) IRM method, yields the least variance of invariant prediction accuracies and the best average prediction accuracy, compared to both IRM-GAME and FISHR. To summarize, the new evaluation method, with diverse test environments, enables us to make a fair comparison of IRM methods implemented in different training environment settings. Unless specified otherwise, we use the multi-environment evaluation method throughout this work.

## 5 ADVANCING IRM-GAME VIA CONSENSUS-CONSTRAINED BLO

In this section, we revist and advance a special IRM variant, IRM-GAME (Ahuja et al., 2020), which endows each individual environment with a separate prediction head and converts IRM into an ensemble game over these multiple predictors.

**Revisiting IRM-GAME.** We first introduce the setup of IRM-GAME following notations used in Section 2. The most essential difference between IRM-GAME and the vanilla IRM framework is that the former assigns each environment with an individual classifier $\mathbf{w}^{(e)}$, and then relies on the ensemble of these individual predictors, i.e., $\frac{1}{N}\sum_{e\in\mathcal{E}_{\mathrm{tr}}}(\mathbf{w}^{(e)}\circ\boldsymbol{\theta})$, for inference. IRM-GAME is in a sharp contrast to IRM, where an environment-agnostic prediction head $\mathbf{w}^*$ simultaneously optimizes the losses across all environments. Therefore, we raise the following question: *Can* IRM-GAME *learn an invariant predictor?*

Inspired by the above question, we explicitly enforce invariance by imposing a *consensus prediction constraint* $\mathcal{C} := \{(\bar{\mathbf{w}}^{(1)}, \bar{\mathbf{w}}^{(2)}, \dots \bar{\mathbf{w}}^{(N)}) \mid \bar{\mathbf{w}}^{(1)} = \dots = \bar{\mathbf{w}}^{(N)}\}$ and integrate it with IRM-GAME. Here, $\bar{\mathbf{w}}^{(e)}$ denotes the prediction head for the $e$-th environment. Based on the newly-introduced constraint, the ensemble prediction head $\frac{1}{N}\sum_{e\in\mathcal{E}_{\mathrm{tr}}}\mathbf{w}^{(e)}$ can be interpreted as the *average consensus* over $N$ environments: $\mathbf{w}^* := \frac{1}{N}\sum_{e\in\mathcal{E}_{\mathrm{tr}}}\mathbf{w}^{(e)} = \arg\min_{\{\bar{\mathbf{w}}^{(e)}\}_e\in\mathcal{C}}\sum_{e\in\mathcal{E}_{\mathrm{tr}}}\|\bar{\mathbf{w}}^{(e)} - \mathbf{w}^{(e)}\|_2^2$. With the above consensus interpretation, we can then cast the invariant predictor-baked IRM-GAME as a consensus-constrained BLO problem, extended from (IRM):

$$\begin{aligned}
\underset{\boldsymbol{\theta}}{\text{minimize}} \quad & \textstyle\sum_{e\in\mathcal{E}_{\mathrm{tr}}}\ell^{(e)}(\mathbf{w}^*(\boldsymbol{\theta})\circ\boldsymbol{\theta}) \\
\text{subject to} \quad & \textbf{(I)}: \mathbf{w}^{(e)}(\boldsymbol{\theta}) \in \underset{\bar{\mathbf{w}}^{(e)}}{\arg\min}\,\ell^{(e)}(\bar{\mathbf{w}}^{(e)}\circ\boldsymbol{\theta}),\ \forall e\in\mathcal{E}_{\mathrm{tr}}, \\
& \textbf{(II)}: \mathbf{w}^*(\boldsymbol{\theta}) = \tfrac{1}{N}\textstyle\sum_{e\in\mathcal{E}_{\mathrm{tr}}}\mathbf{w}^{(e)}(\boldsymbol{\theta}).
\end{aligned} \tag{1}$$

The above contains two lower-level problems: **(I)** per-environment risk minimization, and **(II)** projection onto the consensus constraint ($\{\mathbf{w}^{(e)}\} \in \mathcal{C}$). The incorporation of (II) is intended to ensure the use of invariant prediction head $\mathbf{w}^*(\boldsymbol{\theta})$ in the upper-level optimization problem of (1).

**Limitation of (1) and BLOC-IRM.** In (1), the introduced *consensus-constrained* lower-level problem might compromise the optimality of the lower-level solution $\mathbf{w}^*(\boldsymbol{\theta})$ to the per-environment (unconstrained) risk minimization problem **(I)**, i.e., violating the per-environment *stationarity*

$\|\nabla_{\mathbf{w}} \ell^{(e)}(\mathbf{w}^*(\boldsymbol{\theta}) \circ \boldsymbol{\theta})\|_2^2$. Figure A3 justifies this side effect. As we can see, the per-environment stationarity is hardly attained at the consensus prediction when solving (1). This is not surprising since a constrained optimization solution might not be a stationary solution to minimizing the (unconstrained) objective function. To alleviate this limitation, we improve (1) by explicitly promoting the per-environment stationarity $\|\nabla_{\mathbf{w}} \ell^{(e)}(\mathbf{w}^*(\boldsymbol{\theta}) \circ \boldsymbol{\theta})\|_2^2$ in its upper-level problem through optimization over $\boldsymbol{\theta}$. This leads to **BLOC-IRM** (**BLO** with **C**onsensus **IRM**):

$$\begin{array}{ll} \underset{\boldsymbol{\theta}}{\text{minimize}} & \sum_{e \in \mathcal{E}_{\text{tr}}} \left[ \ell^{(e)}(\mathbf{w}^*(\boldsymbol{\theta}) \circ \boldsymbol{\theta}) + \gamma \|\nabla_{\mathbf{w}} \ell^{(e)}(\mathbf{w}^*(\boldsymbol{\theta}) \circ \boldsymbol{\theta})\|_2^2 \right] \\ \text{subject to} & \text{Lower-level problems \textbf{(I)} and \textbf{(II)} in (1),} \end{array} \qquad \text{(BLOC-IRM)}$$

where $\gamma > 0$ is a regularization parameter like IRMv0. Assisted by the (upper-level) prediction stationarity regularization, the consensus prediction **(II)** indeed simultaneously minimizes the risks of all the environments, supported by the empirical evidence that the convergence of $\|\nabla_{\mathbf{w}} \ell^{(e)}(\mathbf{w}^*(\boldsymbol{\theta}) \circ \boldsymbol{\theta})\|_2^2$ towards 0 along each environment's optimization path (see Figure A3).

Further, we elaborate on how the BLOC-IRM problem can be effectively solved using an ordinary BLO solver. First, it is worth noting that although both (IRM) and BLOC-IRM are BLO problems, the latter is easier to solve since the lower-level constraint **(I)** is unconstrained and separable over environments, and the consensus operation **(II)** is linear. Based on these characteristics, the implicit gradient $\frac{d\mathbf{w}^*(\boldsymbol{\theta})}{d\boldsymbol{\theta}}$ can be directly computed as

$$\frac{d\mathbf{w}^*(\boldsymbol{\theta})}{d\boldsymbol{\theta}} = \frac{1}{N} \sum_{e \in \mathcal{E}_{\text{tr}}} \frac{d\mathbf{w}^{(e)}(\boldsymbol{\theta})}{d\boldsymbol{\theta}}, \quad \text{subject to } \mathbf{w}^{(e)}(\boldsymbol{\theta}) \in \underset{\bar{\mathbf{w}}^{(e)}}{\arg\min} \ell^{(e)}(\bar{\mathbf{w}}^{(e)} \circ \boldsymbol{\theta}). \qquad (2)$$

Since the above lower-level problem is *unconstrained*, we can call the standard $\arg\min$ differentiating method, such as implicit function approach (Gould et al., 2016) or gradient unrolling (Liu et al., 2021) to compute $\frac{d\mathbf{w}^{(e)}(\boldsymbol{\theta})}{d\boldsymbol{\theta}}$. In our work, we adopt the gradient unrolling method, which approximates $\mathbf{w}^{(e)}(\boldsymbol{\theta})$ by a $K$-step gradient descent solution, noted by $\mathbf{w}_K^{(e)}(\boldsymbol{\theta})$ and then leverages automatic differentiation (AD) to compute the derivative from $\mathbf{w}_K^{(e)}(\boldsymbol{\theta})$ to the variable $\boldsymbol{\theta}$. Figure 3 shows the working pipeline of BLOC-IRM and its comparison to original IRM and

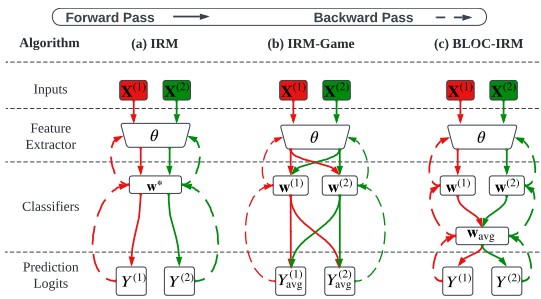

Figure 3: Schematic overview of BLOC-IRM over two training environments (red and green), and its comparison to IRM and IRM-GAME.

IRM-GAME methods. We use $K = 1$ for the lower-level problem throughout our experiments. We refer readers to Appendix B.2 for more algorithmic details. We also explore the performance of our proposed BLOC-IRM with various regularization terms, based on the penalties used in the existing literature. We show the best performance is always achieved when the stationarity is penalized in the upper-level (see Table A3).

## 6 EXPERIMENTS

In this section, we begin by introducing some key experiment setups (with details in Appendix C.1), and then empirically show the effectiveness of our proposed IRM training and evaluation improvements over existing IRM methods across various datasets, models, and learning environments.

### 6.1 EXPERIMENT SETUPS

**Datasets and models.** Our experiments are conducted over 7 datasets as referenced and shown in Tables 1, 2. Among these datasets, COLORED-MNIST, COLORED-FMNIST, CIFAR-MNIST, and COLORED-OBJECT are similarly curated, mimicking the pipeline of COLORED-MNIST (Arjovsky et al., 2019), by introducing an environment bias parameter (*e.g.*, $\beta$ for COLORED-MNIST in Section 4) to customize the level of spurious correlation (as shown in Table 2) in different environments. In the CELEBA dataset, we choose the face attribute 'smiling' (vs. 'non-smiling') as the *core* feature aimed for classification, and regard another face attribute 'hair color' ('blond' vs. 'dark') as the

source of spurious correlation imposed on the core feature. By controlling the level of spurious correlation, we then create different training/testing environments in CELEBA. Furthermore, we study PACS and VLCS datasets, which were used to benchmark domain generalization ability in the real world (Borlino et al., 2021). It was recently shown by Gulrajani & Lopez-Paz (2020) that for these datasets, ERM could even be better than IRMv1. Yet, we will show that our proposed BLOC-IRM is a promising domain generalization method, which outperforms all the IRM baselines and ERM in practice. In addition, we follow Arjovsky et al. (2019) in adopting multi-layer perceptron (MLP) as the model for resolving COLORED-MNIST and COLORED-FMNIST problems. In the other more complex datasets, we use the ResNet-18 architecture (He et al., 2016).

**Baselines and implementation.** Our baselines include 7 IRM variants (Table 1) and ERM, which are implemented using their official repositories if available (see Appendix C.2). Unless specified otherwise, our training pipeline uses the small-batch training setting. By default, we use the batch size of 1024 for COLORED-MNIST and COLORED-FMNIST, and 256 for other datasets. In Section 6.2 below, we also do a thorough comparison of large-batch vs small-batch IRM training.

**Evaluation setup.** As proposed in Section 4, we use the multi-environment evaluation metric unless specified otherwise. To capture both the accuracy and variance of invariant predictions across multiple testing environments, the *average accuracy* and the *accuracy gap* (the difference of the best-case and worst-case accuracy) are measured for IRM methods. The resulting performance is reported in the form $a \pm b$, with mean $a$ and standard deviation $b$ computed across 10 independent trials.

## 6.2 EXPERIMENT RESULTS

**Small-batch training improves all existing IRM methods on COLORED-MNIST & COLORED-FMNIST.** Recall from Section 3 that all the existing IRM methods (Table 1) adopt full-batch IRM training on COLORED-MNIST & COLORED-FMNIST, which raises the large-batch training problem. In **Table 4**, we conduct

Table 4: Performance of existing IRM methods in large and small-batch settings. GRAYSCALE refers to ERM on uncolored data, which yields the best prediction (supposing no spurious correlation during training). The IRM performance is evaluated by average accuracy ('Avg Acc') and accuracy gap ('Acc Gap'), in the format mean±std. A higher Avg Acc and lower Acc Gap is preferred. The theoretically optimal performance is 75% (Arjovsky et al., 2019).

| Dataset | | COLORED-MNIST | | COLORED-FMNIST | |
| --- | --- | --- | --- | --- | --- |
| Metrics(%) | | Avg Acc (↑) | Acc Gap (↓) | Avg Acc (↑) | Acc Gap (↓) |
| GRAYSCALE | | 73.39±0.16 | 0.32±0.03 | 74.05±0.09 | 0.13±0.04 |
| ERM | | 49.19±1.89 | 90.72±2.08 | 49.77±1.71 | 88.62±2.49 |
| IRMv1 | Large Batch | 67.13 ±0.33 | 3.43±0.14 | 67.19±0.22 | 3.35±0.11 |
| IRMv0 | | 65.39 ±0.34 | 4.69±0.18 | 66.44±0.28 | 3.53±0.13 |
| IRM-GAME | | 65.69 ±0.42 | 8.75±0.14 | 65.91±0.29 | 3.74±0.09 |
| REX | | 67.42 ±0.29 | 3.76±0.07 | 67.82±0.31 | 3.26±0.16 |
| BIRM | | 67.93 ±0.31 | 3.81±0.11 | 67.75±0.26 | 3.81±0.11 |
| SPARSEIRM | | 67.72 ±0.28 | 3.65±0.08 | 67.89±0.30 | 3.12±0.15 |
| FISHR | | 67.49±0.39 | 4.37±0.10 | 67.33±0.24 | 4.49±0.16 |
| IRMv1 | Small Batch | 68.33±0.31 | 2.04±0.05 | 68.76±0.31 | 1.45±0.09 |
| IRMv0 | | 68.37±0.28 | 1.32±0.09 | 69.07±0.27 | 1.36±0.06 |
| IRM-GAME | | 67.73±0.24 | 1.67±0.14 | 67.49±0.32 | 1.82±0.13 |
| REX | | 68.42±0.29 | 1.65±0.07 | 68.66±0.22 | 1.29±0.08 |
| BIRM | | 68.71±0.21 | 1.35±0.09 | 68.64±0.32 | 1.44±0.13 |
| SPARSEIRM | | 68.81±0.25 | 1.72±0.05 | 68.29±0.22 | 1.28±0.15 |
| FISHR | | 68.69±0.19 | 2.13±0.08 | 68.79±0.17 | 1.77±0.10 |

a thorough comparison between the originally-used full-batch IRM methods and their small-batch counterparts. In addition, we present the performance of ERM and ERM-grayscale (we call it 'grayscale'), where the latter is ERM on *uncolored* data. In the absence of any spurious correlation in the training set, grayscale gives the *best* performance. As discussed in Section 4 & 6.1, the IRM performance is measured by the average accuracy and the accuracy gap across 19 testing environments, parameterized by the environment bias parameter $\beta \in \{0.05, \dots, 0.95\}$. We make some key observations from Table 4. **First**, small batch size helps improve *all* the existing IRM methods consistently, evidenced by the $1\% \sim 3\%$ improvement in average accuracy. **Second**, the small-batch IRM training significantly reduces the variance of invariant predictions across different testing environments, evidenced by the decreased accuracy gap. This implies that the small-batch IRM training can also help resolve the limitation of multi-environment evaluation for the existing IRM methods, like the sensitivity of IRM-GAME accuracy to $\beta$ in Figure 2. **Third**, we observe that IRMv0, which does not seem to be useful in the large batch setting, becomes quite competitive with the other baselines in the small-batch setting. Thus, large-batch could suppress the IRM performance for some methods. In the rest of the experiments, we stick to the small-batch implementation of IRM training.

**BLOC-IRM outperforms IRM baselines in various datasets.** Next, **Table 5** demonstrates the effectiveness of our proposed BLOC-IRM approach versus ERM and existing IRM baselines across all the 7 datasets listed in Table 2. Evidently, BLOC-IRM yields a higher average accuracy compared to all the baselines, together with the smallest accuracy gap in most cases. Additionally, we observe that CELEBA, PACS and VLCS are much more challenging datasets for capturing invariance through IRM, as evidenced by the small performance gap between ERM and IRM methods. In

Table 5: IRM performance comparison between BLOC-IRM and other baselines. We use ResNet-18 (He et al., 2016) for all the datasets. The evaluation setup is consistent with Table 4, and the best performance per-dataset is highlighted in **bold**. We present the results with the full dataset list in Table A1.

| Algorithm | COLORED-OBJECT | | CIFAR-MNIST | | CELEBA | | VLCS | | PACS | |
|---|---|---|---|---|---|---|---|---|---|---|
| Metrics (%) | Avg Acc | Acc Gap | Avg Acc | Acc Gap | Avg Acc | Acc Gap | Avg Acc | Acc Gap | Avg Acc | Acc Gap |
| ERM | 41.11±1.44 | 86.43±2.89 | 40.39±1.32 | 85.53±2.33 | 72.38±0.29 | 10.73±0.36 | 63.23±0.23 | 12.39±0.35 | 69.95±0.35 | 14.32±0.75 |
| IRMv1 | 64.42±0.21 | 4.18±0.29 | 61.49±0.29 | 7.17±0.33 | 72.49±0.38 | 10.15±0.27 | 62.72±0.29 | 12.74±0.27 | 68.93±0.33 | 14.99±0.51 |
| IRMv0 | 62.39±0.25 | 5.36±0.31 | 60.14±0.18 | 8.83±0.39 | 72.42±0.35 | 10.43±0.38 | 62.59±0.32 | 12.99±0.36 | 68.72±0.29 | 15.29±0.71 |
| IRM-GAME | 62.88±0.34 | 5.59±0.28 | 60.44±0.31 | 6.72±0.41 | 72.18±0.44 | 12.32±0.41 | 62.31±0.38 | 13.37±0.62 | 68.12±0.22 | 15.77±0.66 |
| REX | 63.37±0.35 | 5.42±0.31 | 62.32±0.24 | 5.55±0.32 | 72.34±0.26 | 10.31±0.23 | 63.19±0.31 | 12.87±0.31 | 69.43±0.34 | 15.31±0.67 |
| BIRM | 65.11±0.27 | **3.31**±0.22 | 62.99±0.35 | 5.23±0.36 | 72.93±0.28 | 9.92±0.33 | 63.33±0.40 | 12.13±0.23 | 69.34±0.25 | 15.76±0.49 |
| SPARSEIRM | 64.97±0.39 | 3.97±0.25 | 62.16±0.29 | **4.14**±0.31 | 72.42±0.33 | 9.79±0.21 | 62.86±0.26 | 12.79±0.35 | 69.52±0.39 | 15.81±0.82 |
| FISHR | 64.07±0.23 | 4.41±0.29 | 61.79±0.25 | 5.55±0.21 | 72.89±0.25 | 9.42±0.32 | 63.44±0.37 | 11.93±0.42 | 70.21±0.22 | **14.52**±0.43 |
| BLOC-IRM | **65.97**±0.33 | 4.10±0.36 | **63.69**±0.32 | 4.89±0.36 | **73.35**±0.32 | **8.79**±0.21 | **63.62**±0.35 | **11.55**±0.32 | **70.31**±0.21 | 14.73±0.65 |

particular, all the IRM methods, except FISHR and BLOC-IRM, could even be worse than ERM on PACS and VLCS. Here, we echo and extend the findings of Krueger et al. (2021, Section 4.3). However, we also show that BLOC-IRM is a quite competitive IRM variant when applied to realistic domain generalization datasets. We also highlight that the CELEBA experiment is newly constructed and performed in our work for invariance evaluation. Like PACS and VLCS, this experiment also shows that ERM is a strong baseline, and among IRM-based methods, BLOC-IRM is the best-performing, both in terms of accuracy and variance of invariant predictions.

**IRM against model size and training environment variation.** Furthermore, we investigate the effect of model size and training environment diversity on the IRM performance. The recent works (Lin et al., 2022; Zhou et al., 2022b) have empirically shown that IRMv1 may suffer a significant performance loss when trained over large-sized neural network models, and thus developed BIRM and SPARSEIRM approaches as advancements of IRMv1. Inspired by these works, **Figure 4** presents the sensitivity of invariant prediction to model size for different IRM methods on COLORED-MNIST. Here the model size is controlled by the dimension of the intermediate layer (denoted by $d$) in MLP, and the default dimension is $d = 390$ (*i.e.*, the vertical dotted line in Figure 4), which was used in (Arjovsky et al., 2019) and followed in the subsequent literature. As we can see, when $d > 390$, nearly all the studied IRM methods (including BLOC-IRM) suffer a performance drop. Yet, as $d \geq 800$, from the perspective of prediction accuracy and model resilience together, the top-3 best IRM methods with model size resilience are BIRM, SPARSEIRM, and BLOC-IRM, although we did not intentionally design BLOC-IRM to resist performance degradation against model size.

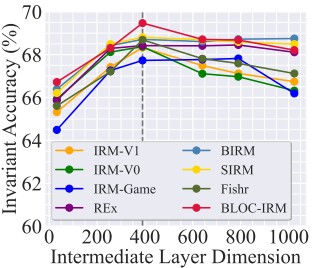

Figure 4: IRM performance on COLORED-MNIST against the layer dimension in MLP. The dotted line represents the default dimension ($d = 390$) used in the literature. The invariant prediction accuracy is presented via the dot line (mean). The results are based on 10 independent trials and we report the variance in Figure A4.

We also show more experiment results in the Appendix. In **Table A2**, we study IRM with different numbers of training environment configurations and observe the consistent improvement of BLOC-IRM over other baselines. In **Table A4** we show that the performance of invariant prediction degrades, if additional covariate shifts (class, digit, and color imbalances on COLORED-MNIST) are imposed on the training environments following Krueger et al. (2021, Section 4.1) and also demonstrate that BLOC-IRM maintains the accuracy improvement over baselines with each variation. In **Table A5**, we compare the performance of different methods in the failure cases of IRM pointed out by (Kamath et al., 2021) and show the consistent improvement brought by BLOC-IRM.

## 7 CONCLUSION

In this work, we investigate existing IRM methods and reveal long-standing but chronically overlooked challenges involving IRM training and evaluation, which may lead to sub-optimal solutions and incomplete invariance assessment. As a remedy, we propose small-batch training and multi-environment evaluation. We reexamine the IRM-GAME method through the lens of consensus-constrained BLO, and develop a novel IRM variant, termed BLOC-IRM. We conducted extensive experiments on 7 datasets and demonstrate that BLOC-IRM consistently improves all baselines.

## ACKNOWLEDGEMENT

The work of Y. Zhang and S. Liu was partially supported by National Science Foundation (NSF) Grant IIS-2207052. The work of M. Hong was supported by NSF grants CNS-2003033 and CIF-1910385. The computing resources used in this work were partially supported by the MIT-IBM Watson AI Lab and the Institute for Cyber-Enabled Research (ICER) at Michigan State University.

## REPRODUCIBILITY STATEMENT

The authors have made an extensive effort to ensure the reproducibility of algorithms and results presented in the paper. First, the details of the experiment settings have been elaborated in Section 6.1 and Appendix C.1. In this paper, seven datasets are studied and the environment generation process for each dataset is described with details in Appendix A. The evaluation metrics are also clearly introduced in Section 3. Second, eight IRM-oriented methods (including our proposed BLOC-IRM) are studied in this work. The implementation details of all the baseline methods are clearly presented in Appendix C.2, including the hyper-parameters tuning, model configuration, and used code bases. For our proposed BLOC-IRM, we include all the implementation details in Section 5 and Appendix B.2, including training pipeline in Figure 3 and the pseudo-code in Algorithm A1. Third, all the results are based on 10 independent trials with different random seeds. The standard deviations are also reported to ensure fair comparisons across different methods. Fourth, codes are available at `https:/github.com/OPTML-Group/BLOC-IRM`.

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

# APPENDIX

## A DATASET SELECTION

Compared to existing work, we expand the dataset types for evaluating the performance of different IRM methods (see Table 2). In addition to the most commonly-used benchmark datasets COLORED-MNIST (Arjovsky et al., 2019) and COLORED-FMNIST (Ahuja et al., 2020), we also consider the datasets CIFAR-MNIST (Lin et al., 2021; Shah et al., 2020) and COLORED-OBJECT (Ahmed et al., 2020; Zhang et al., 2021), which impose artificial spurious correlations, MNIST digit number and object color, into the original CIFAR-10 and COCO Detection datasets, respectively. Furthermore, we consider other three real-world datasets CELEBA (Liu et al., 2015), PACS (Li et al., 2017) and VLCS (Torralba & Efros, 2011), without imposing artificial spurious correlations. Notably, CELEBA was first formalized and introduced to benchmark IRM performance. The recent work (Gulrajani & Lopez-Paz, 2020) showed that when carefully implemented, ERM could outperform IRMv1 in PACS and VLCS. Thus, we regard them as challenging datasets to capture invariance.

For COLORED-OBJECT dataset, we strictly follow the setting adopted in (Lin et al., 2022) to generate the spurious features. For CIFAR-MNIST we use the class "bird" and "plane" in the dataset `CIFAR` as the invariant feature, while the digit "0" and "1" in `MNIST` as the spurious correlation.

CELEBA dataset is, for the first time, introduced to measure IRM performance. We select the attribute "Smiling" as the invariant label and use the attribute "Hair Color" (blond and black hair) to create a spurious correlation in each environment.

## B IMPLEMENTATION DETAILS

### B.1 DETAILS ON LARGE-BATCH OPTIMIZATION ENHANCEMENTS

✦ **IRM-LSGD**: We first integrate large-batch SGD (LSGD) with IRM. Following (Goyal et al., 2017), we make two main modifications: (1) scaling up learning rate linearly with batch size, and (2) prepending a warm-up optimization phase to IRM training. We call the LSGD-baked IRM variant IRM-LSGD.

✦ **IRM-LALR**: Next, we adopt layerwise adaptive learning rate (LALR) in IRM training. Following (You et al., 2019), we advance the learning rate scheduler by assigning each layer of a neural network-based prediction model with an *adaptive* learning rate (*i.e.*, proportional to the norm of updated model weights per layer). More specifically, the model parameter update rule becomes:

$$\boldsymbol{\theta}_{t+1,i} = \boldsymbol{\theta}_{t,i} - \frac{\tau(\|\boldsymbol{\theta}_{t,i}\|_2^2) \cdot \eta_t}{\|\mathbf{u}_{t,i}\|_2^2} \mathbf{u}_{t,i}, \tag{A1}$$

where $\boldsymbol{\theta}_{t,i}$ denotes the $i$-th layer of the model parameters at iteration $t$, and $\mathbf{u}_{t,i}$ represents the first-order gradient of the corresponding layer-wise model parameters. We use $\tau(\|\boldsymbol{\theta}_{t,i}\|_2^2 = \min\{\max\{\|\boldsymbol{\theta}_{t,i}\|_2^2, c_l\}, c_u\})$ as the scaling factor of the adaptive learning rate $\frac{\eta_t}{\|\mathbf{u}_{t,i}\|}$. We use $c_l = 0$ and $c_u = 1$ in our experiments.

✦ **IRM-SAM**: Lastly, we leverage sharpness-aware minimization (SAM) to simultaneously minimize the IRM loss and the loss sharpness. The latter is achieved by explicitly penalizing the worst-case training loss of model weights when facing small weight perturbations. This yields a wide minimum within a flat loss landscape. More specifically, the sharpness-aware loss can be formulated as:

$$\min_{\boldsymbol{\theta}} \ell^{\text{SAM}}(\boldsymbol{\theta}), \quad \text{where} \quad \ell^{\text{SAM}}(\boldsymbol{\theta}) = \max_{\|\boldsymbol{\epsilon}\|_2^2 \le \rho} \ell(\boldsymbol{\theta} + \boldsymbol{\epsilon}), \tag{A2}$$

where the parameter perturbation $\boldsymbol{\epsilon}$ is subject to the perturbation constraint $\|\boldsymbol{\epsilon}\|_2^2 \le \rho$. When applied to IRM, we replace the per-environment training loss with the SAM loss, and adopt the $\rho = 0.001$.

### B.2 BLOC-IRM IMPLEMENTATION

As described in Section 5, the BLOC-IRM algorithm solves the IRM problem with two optimization levels. We use 1-step gradient descent to get the lower-level solution. We retain the gradient

graph in PyTorch to enable auto differentiation. We assign each of the classification head $\{\mathbf{w}^{(e)}\}$ a separate optimizer and use the same learning rate as the feature extractor $\boldsymbol{\theta}$. For COLORED-MNIST and COLORED-FMNIST, we adopt a learning rate of $2 \times 10^{-3}$ and use the Adam (Kingma & Ba, 2014) optimizer. As for other datasets, we use the multi-step learning rate scheduler with an initial learning rate of 0.1, which is consistent with other baselines. We adopt the same penalty weight of $10^6$ as IRMv1 and IRMv0.

---

**Algorithm A1** BLOC-IRM

1: **Initialization**: Training data $\{\mathbf{x}^{(e)}\}$ from $N$ environments, Model feature extractor $\boldsymbol{\theta}_0$, and $N$ model classification heads $\{\mathbf{w}_0^{(e)}\}$, learning rate $\{\eta_t\}$ series, penalty weight $\{\gamma_t\}$ series.
2: **for** Step $t = 0, 1, \ldots,$ **do**
3:     **Lower-level:** update classification head for each environment:

$$\forall e \in \mathcal{E}_{\mathrm{tr}}, \qquad \tilde{\mathbf{w}}_{t+1}^{(e)} = \mathbf{w}_t^{(e)} - \eta_t \left. \frac{d\ell^{(e)}(\mathbf{w} \odot \boldsymbol{\theta})}{d\mathbf{w}} \right|_{\boldsymbol{\theta}=\boldsymbol{\theta}_t, \mathbf{w}=\mathbf{w}_t^{(e)}} \tag{A3}$$

4:     **Consensus projection:** $\forall e \in \mathcal{E}_{\mathrm{tr}}, \mathbf{w}_{t+1}^{(e)} = \mathbf{w}_{t+1}^* = \frac{1}{N} \sum_{e \in \mathcal{E}_{\mathrm{tr}}} \tilde{\mathbf{w}}_{t+1}^{(e)}$
5:     **Upper-level:** update feature extractor with stationary penalty:

$$\boldsymbol{\theta}_{t+1} = \boldsymbol{\theta}_t - \eta_t \sum_{e \in \mathcal{E}_{\mathrm{tr}}} \frac{d}{d\boldsymbol{\theta}} \left( \ell^{(e)}(\mathbf{w} \odot \boldsymbol{\theta}) + \gamma_t \|\nabla_{\mathbf{w}} \ell^{(e)}(\mathbf{w} \circ \boldsymbol{\theta})\|_2^2 \right) \Big|_{\boldsymbol{\theta}=\boldsymbol{\theta}_t, \mathbf{w}=\mathbf{w}_{t+1}^*} \tag{A4}$$

6: **end for**

---

## C    EXPERIMENTATION

### C.1    ENVIRONMENT SETUP

As proposed in Section 4, we use the multi-environment evaluation metric unless specified otherwise. To capture both the accuracy and variance of invariant predictions across multiple testing environments, the *average accuracy* and the *accuracy gap* (the difference between the best-case and worst-case accuracy) are evaluated for IRM methods.

Specifically, for the COLORED-MNIST, COLORED-FMNIST, COLORED-OBJECT, CIFAR-MNIST, and CELEBA dataset, we manually create 19 test environments with uniformly sampled bias parameter $\beta \in \{0.05, 0.1, \ldots, 0.95\}$, where the environment bias parameter $\beta$ controls the spurious correlation (see Section 4 for more details).

For VLCS and PACS datasets, the training and test sets have 4 environments, namely {art painting, cartoon, sketch, photo} and {CALTECH, LABELME, PASCAL, SUN} respectively. We use the first three environments as the training environments, while we use the test set of all four environments to form our proposed multi-environment invariance evaluation system.

### C.2    BASELINES

For each baseline method, we follow its official PyTorch repository except IRM-GAME and SPARSEIRM. We translate the TensorFlow-based original code base of IRM-GAME to PyTorch. As one of the latest IRM advancements, the official code of SPARSEIRM is not yet publicly available. Therefore, we reproduce SPARSEIRM in PyTorch.

In particular, for COLORED-MNIST and COLORED-FMNIST, we stick to the original hyperparameters for the large-batch setting and tune the hyper-parameters of each method, including the penalty weight, number of warm-up epochs, and learning rate for the small batch setting.

In particular, for the large-batch setting, we use the penalty weight of $10^6$, 190 warm-up epochs, and 500 epochs in total, as suggested by the original IRMv1 and inherited by its variants. For the small-batch setting, we adopt the same penalty weight $10^6$. Further, we found that the warm-up

phase could be shortened without sacrificing accuracy. Therefore, we use 50 warm-up epochs and total 200 epochs for all the methods.

For other datasets, we adopt the batch size of 128 and use ResNet-18 as the default model architecture. We train for 200 epochs. We adopt the step-wise learning rate scheduler with an initial learning rate of 0.1. The learning rate decays by 0.1 at the 100th and 150th epochs.

## C.3 ADDITIONAL EXPERIMENT RESULTS

**The influence of batch size with all the baselines.** We show in Figure A1 the influence of training batch size on the performance of different methods. We observe in Figure A1, as in Figure 1, that full batch setting does not achieve the best performance, and the use of mini-batch (stochastic gradient descent) indeed improves performance.

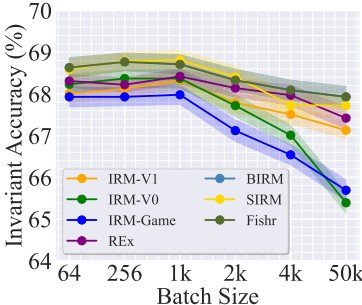

Figure A1: The performance of all the baselines in this work trained with different batch sizes on COLORED-MNIST dataset. The full data batch-size is 50k. The invariant accuracy corresponds to the average accuracy evaluated based on the diversified environments-based evaluation metric.

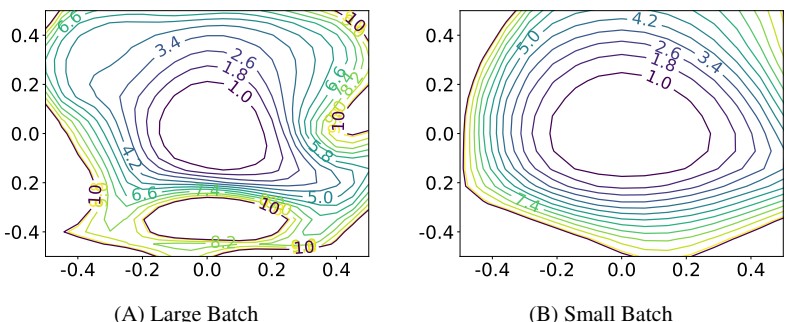

(A) Large Batch    (B) Small Batch

Figure A2: The loss landscapes of invariant prediction models acquired by (A) large-batch IRMv1 training with 50k batch size and (B) small-batch training with 1k batch size. The 2D loss landscape visualization is realized using the tool in (Li et al., 2018). The $x$ and $y$ axes represent the linear interpolation coefficients over two directional vectors originated from the converged local optima. Here the numbers on the contour denote the loss values over test data.

**Loss landscapes of IRMv1 with different batch sizes.** We plot the loss landscapes of the models trained with IRMv1 on COLORED-MNIST using large (full) and small batch in Figure A2. Using small batch training, IRMv1 (Fig. A2B) converges to a smooth neighborhood of a local optima. This also corresponds to a flatter loss landscape than the landscape of the large-batch training (Figure A2(A)). The loss landscapes demonstrate consistent results as other experiments discussed in Section 3.

**Training trajectory with BLOC-IRM with and without stationary loss.** In Figure A3, we plot the per-environment training trajectory of stationary loss when solving (1) and (BLOC-IRM) on COLORED-MNIST. For (BLOC-IRM) we use the regularization term $\lambda = 10^6$, which is aligned with the penalty coefficient used in IRMv1. As we can see, without the stationarity regularization,

the stationary loss remains at a high level for both environments (the dotted curves). Notably, the lower-level stationary can be reached fast with the stationarity penalty, as shown in the solid curves.

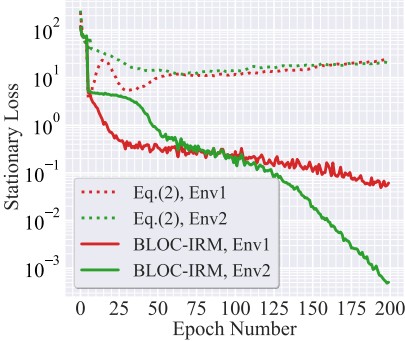

Figure A3: The per-environment training trajectory for the stationarity loss of (1) and (BLOC-IRM) on COLORED-MNIST. The training setting is the same as Figure 2. The algorithmic details can be found in Appendix B.

**Performance of all the methods with full dataset list.** We show in Table A1 the results of all the methods on the seven datasets we studied. To be more specific, in Table A1, we append the results of COLORED-MNIST and COLORED-FMNIST into Table 5 as a whole. As we can see, our methods outperforms other baselines in all the datasets in terms of average accuracy, and stands top in most cases in terms of the accuracy gap.

Table A1: IRM performance comparison between our proposed BLOC-IRM method and other baselines under the full list of datasets. We use MLP for COLORED-MNIST and COLORED-FMNIST, and ResNet-18 (He et al., 2016) for the rest datasets. The evaluation setup is consistent with Table 4, and the best performance per-evaluation metric and per-dataset is highlighted in **bold**.

| Algorithm | COLORED-MNIST | | COLORED-FMNIST | | COLORED-OBJECT | | CIFAR-MNIST | | CELEBA | | VLCS | | PACS | |
| Metrics (%) | Avg Acc | Acc Gap | Avg Acc | Acc Gap | Avg Acc | Acc Gap | Avg Acc | Acc Gap | Avg Acc | Acc Gap | Avg Acc | Acc Gap | Avg Acc | Acc Gap |
|---|---|---|---|---|---|---|---|---|---|---|---|---|---|---|
| ERM | 49.19±1.89 | 90.72±2.08 | 49.77±1.71 | 88.62±2.49 | 41.11±1.44 | 86.43±2.89 | 40.39±1.32 | 85.53±2.33 | 72.38±0.29 | 10.73±0.36 | 63.23±0.23 | 12.39±0.35 | 69.95±0.35 | 14.32±0.75 |
| IRMV1 | 68.33±0.31 | 2.04±0.05 | 68.76±0.31 | 1.45±0.09 | 64.42±0.21 | 4.18±0.29 | 61.49±0.29 | 7.17±0.33 | 72.49±0.38 | 10.15±0.27 | 62.72±0.29 | 12.74±0.27 | 68.93±0.33 | 14.99±0.51 |
| IRMV0 | 68.37±0.28 | 1.32±0.09 | 69.07±0.27 | 1.36±0.06 | 62.39±0.25 | 5.36±0.31 | 60.14±0.18 | 8.83±0.39 | 72.42±0.35 | 10.43±0.38 | 62.59±0.32 | 12.99±0.36 | 68.72±0.29 | 15.29±0.71 |
| IRM-GAME | 67.73±0.24 | 1.67±0.14 | 67.49±0.32 | 1.82±0.13 | 62.88±0.34 | 5.59±0.28 | 60.44±0.31 | 6.72±0.41 | 72.18±0.44 | 12.32±0.41 | 62.31±0.38 | 13.37±0.62 | 68.12±0.22 | 15.77±0.66 |
| REx | 68.42±0.29 | 1.65±0.07 | 68.66±0.22 | 1.29±0.08 | 63.37±0.35 | 5.42±0.31 | 62.32±0.24 | 5.55±0.32 | 72.34±0.26 | 10.31±0.23 | 63.19±0.31 | 12.87±0.31 | 69.43±0.34 | 15.31±0.67 |
| BIRM | 68.71±0.21 | 1.35±0.09 | 68.64±0.32 | 1.44±0.13 | 65.11±0.27 | **3.31±0.22** | 62.99±0.35 | 5.23±0.36 | 72.93±0.28 | 9.92±0.33 | 63.33±0.40 | 12.13±0.23 | 69.34±0.25 | 15.76±0.49 |
| SPARSEIRM | 68.81±0.25 | 1.72±0.05 | 68.29±0.22 | 1.28±0.15 | 64.97±0.39 | 3.97±0.25 | 62.16±0.29 | **4.14±0.31** | 72.42±0.33 | 9.79±0.21 | 62.86±0.26 | 12.79±0.35 | 69.52±0.39 | 15.81±0.82 |
| FISHR | 68.69±0.19 | 2.13±0.08 | 68.79±0.17 | 1.77±0.10 | 64.07±0.23 | 4.41±0.29 | 61.79±0.25 | 5.55±0.21 | 72.89±0.25 | 9.42±0.32 | 63.44±0.37 | 11.93±0.42 | 70.21±0.22 | **14.52±0.43** |
| BLOC-IRM | **69.47±0.24** | **1.04±0.07** | **69.43±0.21** | **1.14±0.11** | **65.97±0.33** | 4.10±0.36 | **63.69±0.32** | 4.89±0.36 | **73.35±0.32** | **8.79±0.21** | **63.62±0.35** | **11.55±0.32** | **70.31±0.21** | 14.73±0.65 |

**Experiment on different model sizes.** We show in Figure A4 the influence of the increasing model size on the performance of different baselines considered in this work. Compared to Figure 4, we report additional standard deviation of the 10 independent trials in Figure A4.

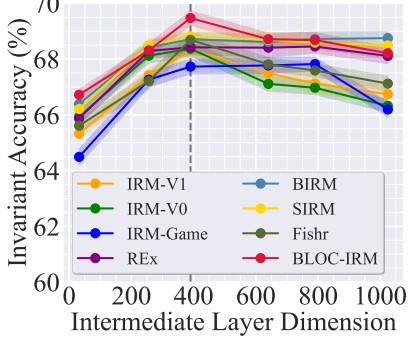

Figure A4: IRM performance on COLORED-MNIST against the dimension of the intermediate layer in MLP. The dotted line represents the default dimension ($d = 390$) used in the literature. The invariant prediction accuracy is presented via the dot line (mean) and shaded area (standard deviation) over 10 random trials.

**Experiment with different training environments.** In Table A2, we show the performance of all the methods in more complex training environments, such as more training environments and more skewed environment bias parameter $\beta$. As we can see, BLOC-IRM outperforms other baselines.

Table A2: Performance under different training environments in COLORED-MNIST.

| Environment Metrics (%) | $p_{\text{tr}} \in \{0.1, 0.15\}$ Avg Acc | Acc Gap | $p_{\text{tr}} \in \{0.1, 0.15, 0.2\}$ Avg Acc | Acc Gap |
|---|---|---|---|---|
| OPTIMUM | 75.00 | 0.00 | 75.00 | 0.00 |
| GRAYSCALE | 73.82±0.11 | 0.37±0.05 | 73.97±0.14 | 0.29±0.08 |
| ERM | 49.21±0.79 | 91.88±3.31 | 49.03±0.93 | 92.17±3.04 |
| IRMV1 | 67.36±0.31 | 2.77±0.15 | 67.11±0.34 | 2.42±0.12 |
| IRMV0 | 67.01±0.42 | 2.85±0.18 | 66.71±0.42 | 2.36±0.19 |
| IRM-GAME | 66.39±0.72 | 4.47±0.61 | 65.93±0.53 | 4.25±0.84 |
| REX | 66.82±0.44 | 2.59±0.11 | 67.14±0.38 | 2.16±0.13 |
| BIRM | 67.35±0.39 | 2.65±0.10 | 68.05±0.43 | **1.99**±0.07 |
| SPARSEIRM | 67.12±0.53 | 2.33±0.18 | 67.72±0.41 | 2.11±0.19 |
| FISHR | 67.22±0.43 | 2.44±0.15 | 67.32±0.39 | 2.59±0.15 |
| BLO-IRM | **68.72**±0.41 | **2.19**±0.15 | **68.89**±0.31 | 2.39±0.09 |

**BLOC-IRM with different regularizations.** Based on the penalty terms used in the existing IRM variants, we explore the performance of our proposed BLOC-IRM with various regularization, including the ones used in IRMV1 (BLOC-IRM-V1), REX (BLOC-IRM-REX), and FISHR (BLOC-IRM-FISHR). We conduct experiments on three different datasets and the results are shown in Table A3. It is obvious that the best performance is always achieved when the per-environment stationarity is penalized in the upper-level. This is not surprising since without an explicit promotion of stationarity, other forms of penalties do not guarantee the BLO algorithm to achieve an optimal solution.

Table A3: The performance of BLOC-IRM with different regularization terms. Three datasets are studied and the latest baseline SPARSEIRM is listed as reference for comparison. The best performance per-evaluation metric and per-dataset is highlighted in **bold**.

| Dataset Metrics | COLORED-MNIST Avg Acc | Acc Gap | COLORED-OBJECT Avg Acc | Acc Gap | CIFAR-MNIST Avg Acc | Acc Gap |
|---|---|---|---|---|---|---|
| SPARSEIRM | 68.81±0.25 | 1.72±0.05 | 64.97±0.39 | **3.97**±0.25 | 62.87±0.29 | **4.14**±0.31 |
| BLOC-IRM | **69.47**±0.24 | **1.04**±0.07 | **65.97**±0.33 | 4.10±0.36 | **63.69**±0.32 | 4.89±0.36 |
| BLOC-IRM-V1 | 67.14±0.24 | 4.33±0.83 | 63.38±0.29 | 6.31±0.51 | 61.13±0.51 | 6.71±0.41 |
| BLOC-IRM-REX | 62.71±0.21 | 8.74±1.21 | 60.31±0.33 | 7.62±0.66 | 59.39±0.55 | 7.89±0.45 |
| BLOC-IRM-FISHR | 63.25±0.16 | 7.12±0.39 | 61.17±0.34 | 6.98.±0.45 | 60.86±0.51 | 6.63±0.30 |

**Performance comparison of different methods with additional covariate shifts.** Besides the sensitivity check on model size, **Table A4** examines the resilience of IRM to variations in the training environment. This study is motivated by Krueger et al. (2021), who empirically showed that the performance of invariant prediction degrades if additional covariate shifts are imposed on the training environments. Thus, we present the IRM performance on COLORED-MNIST by introducing class, digit, and color imbalances, following Krueger et al. (2021, Section 4.1). Compared with Table 4, IRM suffers a greater performance loss in Table A4, in the presence of training environment variations. However, the proposed BLOC-IRM maintains the accuracy improvement over baselines with each variation. In Table A2, we also study IRM with different numbers of training environments and observe the consistent improvement of BLOC-IRM over other baselines.

**Exploration on the failure cases of previous IRM methods.** Some papers either theoretically (Rosenfeld et al., 2020) or empirically (Kamath et al., 2021) pointed out that the original IRMV1 method could fail in certain circumstances, due to the fact that the regularization term used in IRMV1 heavily relies on the "linear predictor" assumption. Regarding this issue, we first bring to attention that the BLOC-IRM formulation does not require the predictors to be linear, since we adopt the regularization in the form of IRMV0 in the upper-level objective, not IRMV1. To justify our argument, we repeat the experiments in (Kamath et al., 2021), which points out a specific scenario using the COLORED-MNIST dataset where IRMV1 fails.

Table A4: IRM performance on COLORED-MNIST and COLORED-FMNIST with training environment variations in terms of class, digit and color imbalances. The best IRM performance per-evaluation metric and per-variation source is highlighted in **bold**.

| Dataset | COLORED-MNIST | | | | | | COLORED-FMNIST | | | | | |
|---|---|---|---|---|---|---|---|---|---|---|---|---|
| Variation | Class Imbalance | | Digit Imbalance | | Color Imbalance | | Class Imbalance | | Digit Imbalance | | Color Imbalance | |
| Metrics (%) | Avg Acc | Acc Gap | Avg Acc | Acc Gap | Avg Acc | Acc Gap | Avg Acc | Acc Gap | Avg Acc | Acc Gap | Avg Acc | Acc Gap |
| GRAYSCALE | 71.23±0.18 | 2.76±0.11 | 70.31±0.21 | 2.79±0.15 | 72.29±0.16 | 2.88±0.14 | 70.15±0.21 | 2.29±0.12 | 69.92±0.15 | 2.72±0.21 | 73.31±0.11 | 1.17±0.23 |
| ERM | 43.72±1.01 | 92.76±1.45 | 45.89±2.82 | 91.65±1.86 | 46.19±2.88 | 90.88±1.69 | 41.72±1.98 | 93.37±2.15 | 42.39±2.39 | 92.23±2.72 | 45.89±0.27 | 91.31±2.27 |
| IRMv1 | 65.39±0.22 | 4.44±0.29 | 64.89±0.26 | 4.19±0.44 | 66.12±0.25 | 3.31±0.29 | 62.49±0.33 | 4.93±0.45 | 61.88±0.23 | 5.54±0.39 | 64.39±0.44 | 3.79±0.33 |
| IRMv0 | 65.01±0.28 | 4.29±0.33 | 65.13±0.25 | 3.87±0.28 | 66.72±0.25 | 3.01±0.44 | 62.78±0.48 | 5.33±0.47 | 61.62±0.29 | 5.29±0.41 | 64.93±0.27 | 3.28±0.31 |
| IRM-GAME | 62.21±0.42 | 6.45±0.35 | 62.10±0.35 | 6.72±0.44 | 61.82±0.65 | 7.78±0.55 | 60.73±0.84 | 6.24±0.43 | 60.79±0.45 | 6.47±0.82 | 64.32±0.42 | 5.73±0.31 |
| REX | **66.45**±0.25 | 3.39±0.28 | 66.23±0.43 | **3.21**±0.20 | 66.99±0.42 | 3.32±0.27 | 64.89±0.36 | 5.78±0.53 | 63.95±0.25 | 4.73±0.62 | 65.87±0.42 | 4.30±0.42 |
| BIRM | 65.73±0.25 | 4.11±0.31 | 65.73±0.88 | 4.49±0.67 | 66.72±0.24 | 3.47±0.25 | 64.39±0.34 | 4.47±0.39 | 63.24±0.39 | **4.54**±0.42 | 65.08±0.31 | 3.80±0.29 |
| SPARSEIRM | 65.32±0.39 | 4.92±0.22 | 64.44±0.36 | 4.85±0.33 | 66.03±0.32 | **2.85**±0.19 | 64.32±0.51 | 4.15±0.36 | 62.97±0.35 | 5.75±0.52 | 64.72±0.46 | 3.99±0.39 |
| FISHR | 66.13±0.28 | 3.99±0.32 | 65.87±0.42 | 3.72±0.41 | 65.48±0.21 | 4.49±0.31 | 63.62±0.53 | 5.59±0.35 | 62.47±0.26 | 5.72±0.33 | 65.13±0.32 | 4.44±0.21 |
| BLOC-IRM | 66.32±0.27 | **3.11**±0.22 | **66.41**±0.29 | 3.32±0.25 | **67.25**±0.24 | 3.72±0.27 | **65.99**±0.31 | **3.97**±0.43 | **65.13**±0.31 | 5.11±0.45 | **66.79**±0.26 | **3.72**±0.36 |

More specifically, the models are trained in the training environments $(\alpha, \beta) = (0.1, 0.2)$ and $(0.1, 0.25)$, and evaluated in the test environment $(0.1, 0.9)$. Note that denotes the label flipping rate and represents the environment bias parameter. The results are shown in the Table A5. As we can see, IRMv1 is clearly worse than ERM as it achieves much lower average accuracy and higher accuracy gap. However, BLOC-IRM outperforms ERM by obtaining high average accuracy and lower accuracy gap. This result shows that BLOC-IRM seems promising to address the empirical IRM challenge discovered in (Kamath et al., 2021). In the meantime, we also acknowledge that BLOC-IRM is not per-

Table A5: Performance comparisons on COLORED-MNIST among ERM, IRMv1, and BLOC-IRM in the scenarios where IRM-variants failed following (Kamath et al., 2021).

| Method | Avg. Acc. | Acc. Gap |
|---|---|---|
| ERM | 83.09 | 13.79 |
| IRMv1 | 76.89 | 27.68 |
| BLOC-IRM | 84.22 | 11.01 |

fect since the advantage achieved by BLOC-IRM over ERM is not strong enough. However, we stress that the main contribution of BLOC-IRM does not lie in solving the failure cases of IRMv1, but to fix the issue of IRM-Game that resorts to a predictor ensemble to make the invariant prediction, which deviates from the spirit of acquiring invariant predictors in the original IRM paradigm.

**A similar curve to Figure 1 on COLORED-FMNIST.** We show the results for COLORED-FMNIST similar to Figure 1 in Figure A5and the conclusion does not change much. As mentioned before, the large-batch training setup was typically used for IRM training over the COLORED-MNIST and COLORED-FMNIST datasets.

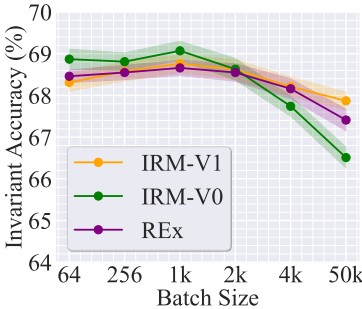

Figure A5: The performance of three IRM methods (IRMv1, IRMv0, and REX) vs. batch size under COLORED-FMNIST.

