# OpenReview forum: "What Is Missing in IRM Training and Evaluation? Challenges and Solutions"
_ICLR.cc/2023/Conference — ICLR 2023 poster_

### Official Review · Reviewer_3u9X · 2022-10-25

**Confidence:** 3
**Correctness:** 3
**Technical Novelty And Significance:** 3
**Empirical Novelty And Significance:** 3
**Recommendation:** 6

**Clarity, Quality, Novelty And Reproducibility:**

The paper is generally clear. I do have a concern about the result in Figure 2a, as it seems odd that the IRM-Game method would have such high accuracy for larger beta, considering that the performance is better than the optimal model using only invariant features (75%), and is also much better than the results shown in their original paper.

The proposed method is novel to the best of my knowledge.

**Strength And Weaknesses:**

Strengths:
- The paper is generally clear and well-written.
- The proposed method outperforms the baselines.

Weaknesses:
1. The paper would be greatly improved with some theoretical justifications of the method. For example, in the original IRM Games paper, they showed that the set of predictors obtained with their method is the same as the one obtained via IRM. Is this also true for BLOC-IRM? In addition, Is the proposed method still susceptible to the failure modes of IRM explored in (Rosenfeld et al., 2020, Kamath et al., 2021)?

2. The authors should apply their method to some real-world shifts from the WILDS benchmark (Koh et al., 2021).

3. Regarding Section 4, though I agree that methods should be evaluated on a diverse set of test environments, I don't necessarily agree with just showing the average performance, as averaging performance over a large grid of test environments can hide performance issues in each one. For example, if most of the test environments are similar to the training environments, then this defeats the purpose of domain generalization. I think perhaps it makes more sense to only consider test environments that are sufficiently different from the training. The authors should also show the worst-environment performance in all the evaluations.

4. The authors should explain why, in Figure 1, the performance with very small batch size (64) is greatly diminished. Is this an issue of compatibility with the other hyperparameters (e.g. learning rate, annealing steps), or is there another justification for this?

5. The authors should show a similar curve to Figure 1 for all of the other datasets (perhaps in the appendix), as this is an interesting and important empirical result.

6. The authors should clarify their model selection method, as this has been found to be extremely important factor in the performance of domain generalization methods (Gulrajani & Lopez-Paz, 2020). Presumably, the authors used the test domain to select their hyperparameters.

7. The proposed method (Section 5) seems disconnected from the previous results, as the method formulation is not motivated by any of empirical findings there.

**Summary Of The Paper:**

The authors first conduct an empirical study of IRM and the IRM Games method, finding that smaller batch sizes can lead to performance gains, and highlighting the importance of evaluating on multiple test environments. They then propose a new variant of IRM where an invariant prediction head is computed as the average of prediction heads from each environment. The authors evaluate their method on several datasets, finding that it outperforms the baselines.

**Summary Of The Review:**

The paper presents a couple of interesting empirical results, as well as proposes a method with solid empirical performance. I am leaning towards accept, pending the authors' rebuttal.

---

### Official Review · Reviewer_U4vc · 2022-10-25

**Confidence:** 4
**Correctness:** 4
**Technical Novelty And Significance:** 3
**Empirical Novelty And Significance:** 3
**Recommendation:** 8

**Clarity, Quality, Novelty And Reproducibility:**

The paper is well-written and clear. I found section 5 to be difficult to read because the formulation there is given a sparse treatment.

**Strength And Weaknesses:**

### Strengths
**Comprehensive empirical demonstration**: for the three changes that this paper proposes to IRM, the authors perform comprehensive assessment of these changes and present empirical evidence that demonstrates improvement in each case. I'll speak to the scale of the improvement in the next section. However, we observe from Table 4 that across the board, small-batch training does indeed improve performance of the IRM variants. Table 5 and 6 show that the BLOC-IRM proposal indeed provides additional gains as well. All the values reported also have error bars as well, so the results should likely hold up to scrutiny.

### Weaknesses
- **BLOC-IRM Formulation**: First the BLO acronym is not really introduced. Second, it is unclear to me why this formulation addresses the challenge discussed in the paper by Rosenfield et. al. on the limitations of IRM. Further, it seems that this formulation would actually not side-step this issue since you need as many classifiers as you have environments.
- **Small batch training**: While small-batch training does give performance improvements, it is unclear why it does so. In addition, these improvements seem quite marginal but consistent. Any thoughts as to why?
- **Unifying theme**: It is unclear how the three portions of this paper fit together.
- **Why IRM-Game**: Why did the authors choose the IRM-Game formulation? It seems like one of out several IRM variants


___
Post Rebuttal Notes
The response has addressed most of my concerns, so I am raising my score.

**Summary Of The Paper:**

This paper proposes improvements to the invariant risk minimization (IRM) framework that ought to make the IRM approach more successful in practice. Specifically, the paper proposes to train IRM classifiers using smaller-batch data, conducts a more comprehensive assessment of current methods to show that some of the previously claimed benefits regarding invariance to environments might not be the case, and lastly introduces a modification of the IRM-Game algorithm of Ahuja et al. Overall, the comprehensive experiments suggests that this approach is beneficial.

**Summary Of The Review:**

This paper proposes 3 different modifications and checks to the current IRM paradigm. Overall, the paper provides comprehensive empirical justification for its claims; however, it is unclear how these modifications fit together.

---

### Official Review · Reviewer_bEZc · 2022-10-27

**Confidence:** 3
**Correctness:** 3
**Technical Novelty And Significance:** 2
**Empirical Novelty And Significance:** 3
**Recommendation:** 6

**Clarity, Quality, Novelty And Reproducibility:**

This paper provides a clear explanation, and the experimental results are concerted.

Besides, they provide implementation code for the reproducibility

**Strength And Weaknesses:**

Strength
1. This paper tackles the critical problems of IRM, and practical solutions to solve the issues.
2. The solutions are simple and effective.
3. The experiment sections are extensive, and they include diverse baselines.

Weakness or Questions
1. This paper provides the Block-IRM, the extension of the IRM game. Is it still possible to analyze the Block-IRM theoretically?
2. This paper extends the IRM as IRM-SAM. I wonder about the author's opinion about the relationship between IRM and sharpness.
3. This paper provides a diverse dataset, but the dataset is limited to the toy or the small-scale dataset. Validation on the large-scale dataset such as DomainNet will be helpful.

**Summary Of The Paper:**

Invariant Risk Minimization (IRM) has recently attracted great attention for its invariance and causality. However, the performance of IRM is limited, and it shows less performance compared to the ERM. This paper raises practical solutions to solve the issues. This paper provides three solutions, batch size, validation set, and new IRM variants. The paper is well-written, and it has a persuasive discussion and experimental results.

**Summary Of The Review:**

I read the paper three times carefully, and the paper is well written and novel.

---

### Decision · Program_Chairs · 2023-01-20

**Decision:**

Accept: poster

**Justification For Why Not Higher Score:**

The paper is well-written and clear with strong experiential results. The only concern is its theoretical justification.

**Justification For Why Not Lower Score:**

N/A

**Metareview: Summary, Strengths And Weaknesses:**

The paper revisits the recent IRM advancements and identifies and resolves three practical limitations in IRM training and evaluation. They find that the effect of batch size during training has been chronically overlooked in previous studies, leaving room for further improvement. They find that improper selection of evaluation environments could give a false sense of invariance for IRM. And they revisit Ahuja et al. (2020)’s proposal to convert IRM into an ensemble game and identify a limitation when a single invariant predictor is desired instead of an ensemble of individual predictors. They conduct experiments (covering 7 existing IRM variants and 7 datasets) to justify the IRM training and evaluation.

++ The paper is generally clear and well-written. The method outperforms the baselines.

-- As raised by the reviewers, the weaknesses of the paper are a lack of theoretical justification of the proposed approach and the phenomena observed by the authors. And reviewer has concerns about the experiment such as the WILDS benchmark (Koh et al., 2021).

The meta-reviewer has carefully read all reviews, the responses, and the paper. The meta-reviewer sees that the authors did a very good job by answering the reviewers' questions. Although the authors cannot complete Camelyon17 in the WILDS benchmark due to the limit on computational resources, they provided other results to justify their paper claims. The authors should carefully modify their paper to include the new results and discussions. The meta-reviewer recommends an acceptance.

**Note From Pc:**

if the above contains the word "oral" or "spotlight" please see: "oral" presentation means -> notable-top-5% and "spotlight" means -> notable-top-25%. As stated in our emails, we are disassociating presentation type from AC recommendations